# SAMPLE-EFFICIENT DIFFUSION-BASED CONTROL OF COMPLEX NONLINEAR SYSTEMS

## ABSTRACT

Complex nonlinear system control faces challenges in achieving sample-efficient, reliable performance. While diffusion-based methods have demonstrated advantages over classical and reinforcement learning approaches in long-term control performance, they are limited by sample efficiency. This paper presents SEDC (Sample-Efficient Diffusion-based Control), a novel diffusion-based control framework addressing three core challenges: high-dimensional state-action spaces, nonlinear system dynamics, and the gap between non-optimal training data and near-optimal control solutions. Our approach introduces a novel control paradigm by architecturally decoupling state-action learning and decomposing dynamics, while a guided self-finetuning process iteratively refines the control policy. These coordinated innovations allow SEDC to achieve 39.5%-47.3% better control accuracy than baselines while using only 10% of the training samples, as validated across multiple complex nonlinear dynamic systems. Our approach represents a significant advancement in sample-efficient control of complex nonlinear systems. The implementation of the code can be found here.

## 1 INTRODUCTION

The control of complex systems plays a critical role across diverse domains, from industrial automation (Baggio et al., 2021) and biological networks (Gu et al., 2015) to robotics (Zhang et al., 2022). Given the challenges in deriving governing equations for empirical systems, data-driven control methods—which design control modules directly based on experimental data collected from the system, bypassing the need for explicit mathematical modeling—have gained prominence for their robust real-world applicability (Baggio et al., 2021; Janner et al., 2022; Ajay et al., 2022; Zhou et al., 2024; Liang et al., 2023; Wei et al., 2024b).

Traditional Proportional-Integral-Derivative (PID) (Li et al., 2006) and Model Predictive Control (MPC) (Schwenzer et al., 2021) methods are limited in complex nonlinear systems. PID controllers struggle with nonlinearities, while MPC's performance depends on model accuracy and is computationally intensive for long-horizon tasks. Data-driven approaches have emerged to address these issues, including supervised learning, reinforcement learning (RL), and diffusion-based methods. Supervised methods like Behavior Cloning (BC) (Pomerleau, 1988) and RL methods like Batch Proximal Policy Optimization (BPPO) (Zhuang et al., 2023) often make myopic, step-by-step decisions, leading to suboptimal outcomes in long-horizon tasks. In contrast, diffusion-based methods (Janner et al., 2022; Ajay et al., 2022; Zhou et al., 2024; Liang et al., 2023; Wei et al., 2024b) treat control as a *global trajectory generation* problem. By generating the entire control plan in a single sample, they achieve comprehensive optimization over the full trajectory, avoiding the pitfalls of iterative methods and enabling superior long-term performance.

The success of diffusion models is dependent on their ability to learn complex trajectory distributions. However, in practice, the available trajectory data is often non-optimal and sparse due to collection under empirical rules or random policies and high operational costs. This presents a key challenge for diffusion-based methods: learning effective control policies from limited and suboptimal data. **First, limited data volume impedes sample-efficient learning in high-dimensional systems.** Existing diffusion-based controllers (Wei et al., 2024a;b; Hu et al., 2025) attempt to directly generate long-term ($T$ steps) state-action trajectories by learning a $T \times (P+M)$-dimensional distribution of system states $y^P$ and control inputs $u^M$. This joint distribution implicitly encodes system dynamics of state

transitions under external control inputs, which often leads to physically inconsistent trajectories when training samples are insufficient. **Second, learning control policies for nonlinear systems remains an open challenge both theoretically and practically.** Traditional analytical methods (Baggio et al., 2021) designed for linear systems fail to perform robustly when applied to nonlinear systems. While diffusion-based approaches (Janner et al., 2022; Ajay et al., 2022; Zhou et al., 2024; Zhong et al., 2025; Hu et al., 2025) employ deep neural networks (e.g., U-Net architectures) as denoising modules to capture nonlinearity, learning effective control policies from limited data remains particularly challenging for complex systems with strong nonlinearity, such as fluid dynamics and power grids. **Third, extracting improved control policies from non-optimal training data poses fundamental difficulties.** Diffusion-based methods (Janner et al., 2022) struggle when training data significantly deviates from optimal solutions. Although recent work (Wei et al., 2024b) introduces reweighting mechanism to expand the solution space during generation, discovering truly near-optimal control policies remains elusive without explicit optimization guidance.

To address these challenges, we propose SEDC (**S**ample-**E**fficient **D**iffusion-based **C**ontrol), a novel diffusion-based framework for learning control policies of complex nonlinear systems with limited, non-optimal data. At its core, SEDC reformulates the control problem as a denoising diffusion process that samples control sequences optimized for reaching desired states while minimizing energy consumption. We then solve the sample efficiency challenge by addressing its three key aspects. To address the curse of dimensionality, we introduce Decoupled State Diffusion (DSD), which simplifies the learning complexity of the generative task. By diffusing only on the more structured state space, rather than the complex joint state-action space, DSD achieves higher sample efficiency. A separate inverse dynamics model is then used to ensure physical consistency. To tackle strong nonlinearity, we propose Dual-Mode Decomposition (DMD) by designing a dual-UNet denoising module with residual connections. This architecture decomposes system dynamics into hierarchical linear and nonlinear components, enabling structured modeling of complex systems. To bridge the gap between non-optimal offline training data and optimal control policies, we introduce the Guided Self-finetuning (GSF) mechanism, which progressively synthesizes guided control trajectories for iterative finetuning, facilitating exploration beyond initial training data and convergence toward near-optimal control strategies.

Our innovation introduces a data-driven framework that dramatically enhances sample-efficiency in controlling high-dimensional nonlinear systems using diffusion models. We demonstrate SEDC's superiority over traditional, reinforcement learning, and diffusion-based methods through experiments on three typical complex nonlinear systems. Our model demonstrates 39.5%-47.3% improvement in control accuracy compared to state-of-the-art baselines while maintaining better balance between accuracy and energy consumption. In sample efficiency experiments, SEDC matches state-of-the-art performance using only 10% of the training samples. We further confirm our method's scalability on a high-dimensional 2D PDE task, its robustness on non-invertible systems, and its competitive computational efficiency, with ablation studies verifying the contribution of each design component.

## 2 RELATED WORK

Data-driven control encompasses various paradigms, which can be broadly categorized by their approach to trajectory generation. One major paradigm is *iterative, feedback-based control*. Classical methods like PID controllers (Li et al., 2006) operate via real-time error correction but face limitations in high-dimensional complex scenarios. More contemporary approaches center on system identification, such as Dynamic Mode Decomposition (Tu, 2013) and Koopman operator theory (Mauroy et al., 2020). These methods first learn an explicit dynamics model from data and then design a controller, often within a Model Predictive Control (MPC) framework (Schwenzer et al., 2021). While robust for real-time adaptation, this two-stage paradigm is susceptible to compounding errors, particularly in sample-scarce settings. Inaccuracies in the learned model can accumulate over long horizons, degrading control performance. Supervised learning (Pomerleau, 1988) and reinforcement learning (Haarnoja et al., 2018; Zhuang et al., 2023) offer adaptive approaches but can also struggle with long-horizon credit assignment and compounding errors.

In contrast, *global trajectory planning* reframes control as a holistic generation problem, for which denoising diffusion models (Ho et al., 2020; Dhariwal & Nichol, 2021; Kong et al., 2020; Ho et al., 2022) have emerged as a powerful tool. By generating the entire control plan as a single,

coherent sample, these methods capture long-term dependencies and avoid the pitfalls of iterative error accumulation. Seminal works (Janner et al., 2022; Ajay et al., 2022) demonstrated this potential in robotics, but their generic architectures struggle with strong nonlinearities; our Dual-Mode Decomposition (DMD) architecture addresses this with a structured inductive bias. Subsequent research has tackled specific limitations. DiffPhyCon (Wei et al., 2024b) uses reweighting to explore beyond the training distribution. This approach, however, requires training separate denoising networks to model decomposed energy functions (one for the prior and one for the conditional distribution). Moreover, its joint state-action modeling can exacerbate the curse of dimensionality, which our Decoupled State Diffusion (DSD) alleviates by diffusing over the state space alone. To bridge the data-optimality gap, AdaptDiffuser (Liang et al., 2023) fine-tunes on discriminator-filtered trajectories, whereas our Guided Self-finetuning (GSF) employs a simpler, filter-free loop. Distinctly, RDM (Zhou et al., 2024) focuses on inference-time adaptation, operating as a replanning framework that corrects trajectories online based on likelihood estimates, rather than improving the generative model offline.

## 3 BACKGROUNDS

### 3.1 PROBLEM SETTING

The dynamics of a controlled complex system can be represented by the differential equation $\dot{\mathbf{y}}_t = \Phi(\mathbf{y}_t, \mathbf{u}_t)$, where $\mathbf{y}_t \in \mathbb{R}^N$ represents the system state and $\mathbf{u}_t \in \mathbb{R}^M$ denotes the control input. We assume the system satisfies the controllability condition without loss of generality: for any initial state $\mathbf{y}_0^*$ and target state $\mathbf{y}_f$, there exists a finite time $T$ and a corresponding control input $\mathbf{u}$ that can drive the system from $\mathbf{y}_0^*$ to $\mathbf{y}_f$. This assumption ensures the technical feasibility of our control objectives. In practical applications, beyond achieving state transitions, we need to optimize the energy consumption during the control process. The energy cost can be quantified using the L2-norm integral of the control input: $J(\mathbf{y}, \mathbf{u}) = \int_0^T |\mathbf{u}(t')|^2 dt'$. Consider a dataset $D = \{\mathbf{u}^{(i)}, \mathbf{y}^{(i)}\}_{i=1}^P$ containing $P$ non-optimal control trajectories, where each trajectory consists of:(1) complete state trajectories $\mathbf{y}^{(i)}$ sampled at fixed time intervals; (2) corresponding control input sequences $\mathbf{u}^{(i)}$. Based on this dataset, our objective is to find the optimal control input trajectory $\mathbf{u}^* \in \mathbb{R}^{T \times M}$ that satisfies:

$$\mathbf{u}^* = \arg\min_{\mathbf{u}} J(\mathbf{y}, \mathbf{u})$$
$$\text{s.t.} \quad \Psi(\mathbf{u}, \mathbf{y}) = 0, \quad \mathbf{y}_0 = \mathbf{y}_0^*, \quad \mathbf{y}_T = \mathbf{y}_f, \tag{1}$$

where $\mathbf{y} \in \mathbb{R}^{T \times N}$ is the corresponding complete state trajectory given $\mathbf{y}_0$ and $\Psi(\mathbf{u}, \mathbf{y}) = 0$. Here, $\Psi(\mathbf{u}, \mathbf{y}) = 0$ represents the system dynamics constraint implicitly defined by dataset $D$. This constraint effectively serves as a data-driven representation of the unknown dynamics equation $\dot{\mathbf{y}}_t = \Phi(\mathbf{y}_t, \mathbf{u}_t)$.

Our key idea is to train a diffusion-based model to directly produce near-optimal control trajectories $\mathbf{u}_{[0:T-1]}$, providing a starting state $\mathbf{y}_0^*$, the target $\mathbf{y}_f$ and optimized by the cost $J$. Next, we summarize the details of the diffusion-based framework.

### 3.2 DIFFUSION MODEL

Diffusion models have become leading generative models, showing exceptional results across image synthesis, audio generation and other applications (Ho et al., 2020; Dhariwal & Nichol, 2021; Song & Ermon, 2019). These models operate by progressively adding noise to sequential data in the forward process and then learning to reverse this noise corruption through a denoising process. We denote that $\mathbf{x}^k$ represents the sequential data at diffusion timestep $k$. In the forward process, a clean trajectory $\mathbf{x}^0$ is progressively corrupted through $K$ timesteps, resulting in a sequence of increasingly noisy versions $\mathbf{x}^1, \mathbf{x}^2, ..., \mathbf{x}^K$. Each step applies a small amount of Gaussian noise: $q(\mathbf{x}^k|\mathbf{x}^{k-1}) = \mathcal{N}(\mathbf{x}^k; \sqrt{1 - \beta^k}\mathbf{x}^{k-1}, \beta^k \mathbf{I})$, where $\beta^k$ is a variance schedule that controls the noise level. With a large enough $K$ we can get $q(\mathbf{x}^K) \approx \mathcal{N}(\mathbf{x}^K; \mathbf{0}, \mathbf{I})$. In the reverse process, the diffusion model learns to gradually denoise the data, starting from pure noise $\mathbf{x}^K$ and working backward to reconstruct the original plausible trajectory $\mathbf{x}^0$. Each denoising step is conditioned on the start and target state: $p_\theta(\mathbf{x}^{k-1}|\mathbf{x}^k, \mathbf{y}_0^*, \mathbf{y}_f) = \mathcal{N}(\mathbf{x}^{k-1}; \mu_\theta(\mathbf{x}^k, k, \mathbf{y}_0^*, \mathbf{y}_f), \mathbf{\Sigma}^k)$, where $\theta$ represents the learnable parameters of the model and $\Sigma^k$ is from a fixed schedule.

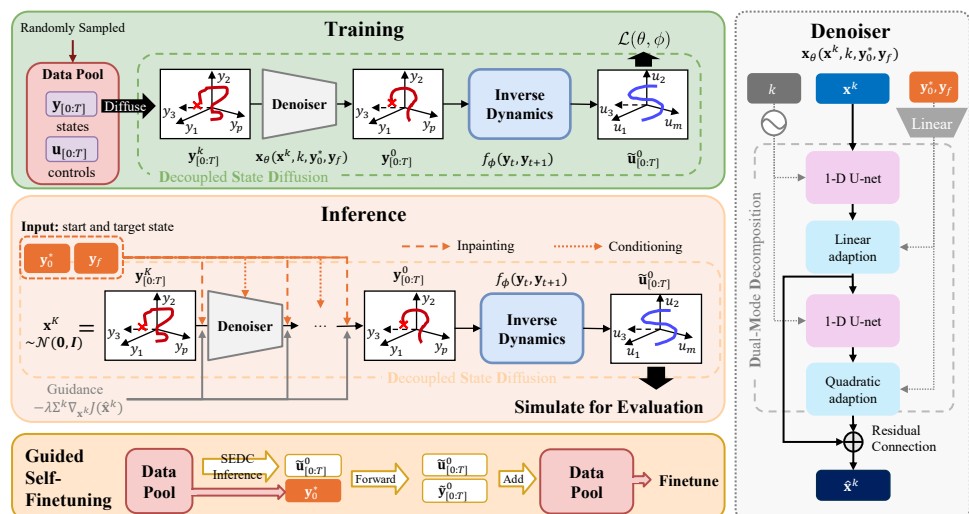

Figure 1: Overview of SEDC. The framework consists of a training/finetuning (top panel), inference process (middle panel) and finetuning process (bottom panel). The core denoising network employs our Dual-Mode Decomposition (DMD) architecture (right panel). Both training and inference leverage Decoupled State Diffusion (DSD) by diffusing only on states and using a separate inverse dynamics model to recover controls.

**Training of diffusion model.** In order to facilitate the design of the denoising network, the network with $\theta$ for the denoising process does not directly predict $\mu$. Instead, it is trained to learn to predict clean trajectory $\mathbf{x}^0$ at every $k$, outputting $\hat{\mathbf{x}}^k$. The training objective for diffusion models typically involves minimizing the variational lower bound (VLB) on the negative log-likelihood (Sohl-Dickstein et al., 2015). In practice, this often reduces to a form of denoising score matching (Song & Ermon, 2019): $\mathbb{E}_{\mathbf{x},k,\mathbf{y}_0^*,\mathbf{y}_f,\epsilon}[||\mathbf{x} - \mathbf{x}_\theta(\mathbf{x}^k, k, \mathbf{y}_0^*, \mathbf{y}_f)||^2]$, where $\mathbf{x}, k, \mathbf{y}_0^*, \mathbf{y}_f$ are sampled from the dataset, $k \sim \mathcal{U}\{1, 2, ..., K\}$ is the step index and $\epsilon \sim \mathcal{N}(\mathbf{0}, \mathbf{I})$ is the noise used to corrupt $\mathbf{x}$.

## 4 SEDC: THE PROPOSED METHOD

### 4.1 CONTROLLING WITH DIFFUSION MODELS

As shown in Figure 1, SEDC reformulates the control problem as a conditional trajectory generation task. The core idea is to train a diffusion model, conditioned on the initial $\mathbf{y}_0^*$ and target $\mathbf{y}_f$ states, to directly generate a complete state trajectory $\mathbf{y}_{[0:T]}$. Subsequently, a corresponding control sequence $\mathbf{u}_{[0:T-1]}$ is derived from this state trajectory via a learned inverse dynamics model.

**Decoupled State Diffusion (DSD).** Jointly modeling the state-action distribution is highly sample-intensive and risks generating physically inconsistent trajectories. To address this, we propose Decoupled State Diffusion (DSD). As shown in Figure 1, we confine the diffusion process to the state trajectory $\mathbf{y}$ alone (i.e., $\mathbf{x} := \mathbf{y}_{[0:T]}$), as state evolution is generally smoother and more structured. The corresponding control sequence is then derived from the generated state transitions via a separately trained inverse dynamics model, $f_\phi$: $\tilde{\mathbf{u}}_{t,\text{update}}^0 = f_\phi(\mathbf{y}_t^0, \mathbf{y}_{t+1}^0)$, where the superscript 0 denotes the final denoised output from the diffusion model. Crucially, the model learns a *plausible control mapping*, a feature that ensures broad applicability without requiring strict system invertibility. It functions by approximating the conditional expectation $\mathbb{E}[\mathbf{u}_t|\mathbf{y}_t, \mathbf{y}_{t+1}]$, which yields an effective average policy even for non-invertible systems where multiple actions could produce the same transition. This decoupled design thus remains robust for a wide range of complex dynamics where an analytical inverse may not exist or be unique.

We then optimize $f_\phi$ simultaneously with the denoiser. The loss function is:

$$\mathcal{L}(\theta, \phi) := \mathbb{E}_{\mathbf{x},k,\mathbf{y}_0^*,\mathbf{y}_f,\epsilon}[||\mathbf{x} - \mathbf{x}_\theta(\mathbf{x}^k, k, \mathbf{y}_0^*, \mathbf{y}_f)||^2] \\ + \mathbb{E}_{\mathbf{y}_t,\mathbf{u}_t,\mathbf{y}_{t+1}}[||\mathbf{u}_t - f_\phi(\mathbf{y}_t, \mathbf{y}_{t+1})||^2], \tag{2}$$

where $\mathbf{y}_t, \mathbf{u}_t, \mathbf{y}_{t+1}$ are sampled from the dataset. After training, the denoising process generates a state trajectory $\mathbf{y}_{0:T}$ that is both physically plausible and conditioned on the start and target states. To further refine this process, we introduce two key mechanisms: inpainting for hard constraint satisfaction and gradient guidance for soft optimization.

**Target-Conditioning via Inpainting.** To ensure the generated trajectory strictly adheres to the given initial state $\mathbf{y}_0^*$ and target state $\mathbf{y}_f$, we treat the problem as a form of trajectory inpainting. While these states are provided as conditions to the denoising network, we enforce them as hard constraints during the sampling process. Specifically, at each denoising step $k$, after sampling a potential trajectory $\mathbf{x}^{k-1} \sim p_\theta(\mathbf{x}^{k-1}|\mathbf{x}^k, \mathbf{y}_0^*, \mathbf{y}_f)$, we replace its start and end points with the ground truth values: $\mathbf{x}_0^{k-1} \leftarrow \mathbf{y}_0^*$ and $\mathbf{x}_T^{k-1} \leftarrow \mathbf{y}_f$. This technique, analogous to inpainting in image generation (Lugmayr et al., 2022), guarantees that the final output satisfies the boundary conditions.

**Cost Optimization via Gradient Guidance.** Beyond satisfying the boundary constraints, we aim to find a trajectory that minimizes a given cost function $J$ (e.g., control energy). We achieve this through inference-time gradient guidance. Building upon the inpainting-enforced sampling, we further modify the mean of the denoising distribution by incorporating the cost gradient:

$$\mu_\theta(\mathbf{x}^k, k, \mathbf{y}_0^*, \mathbf{y}_f) = \frac{\sqrt{\bar{\alpha}^{k-1}}\beta^k}{1-\bar{\alpha}^k}\hat{\mathbf{x}}^k + \frac{\sqrt{\alpha^k}(1-\bar{\alpha}^{k-1})}{1-\bar{\alpha}^k}\mathbf{x}^k$$
$$- \lambda\Sigma^k\nabla_{\mathbf{x}^k}J(\hat{\mathbf{x}}^k(\mathbf{x}^k)), \tag{3}$$

where $\lambda$ controls guidance strength and the superscript $k$ of $\hat{\mathbf{x}}^k := \hat{\mathbf{y}}_{[0:T]}^k$ denotes the clean output from the denoiser at step $k$. Since our diffusion model operates on states only, we recover the control sequence $\tilde{\mathbf{u}}_t^k = f_\phi(\hat{\mathbf{y}}_t^k, \hat{\mathbf{y}}_{t+1}^k)$ to compute the cost $J(\mathbf{u})$ at each step. This combined approach of inpainting and guidance ensures the final generated trajectory is not only feasible and satisfies hard constraints but is also optimized for the desired cost objective.

## 4.2 DUAL-MODE DECOMPOSITION (DMD) FOR DENOISER

In this section, we propose our denoising network design, DMD, that decomposes the modeling of linear and nonlinear modes in the sampled trajectory by a dual-Unet architecture, as shown in Figure 1. Our design draws inspiration from control theory. For linear systems, Yan et al. (2012) demonstrated that optimal control signals have a linear relationship with a specific linear combination $\mathbf{y}_c$ of initial and target states. Building upon this insight, we develop a framework where a bias-free linear layer first learns this crucial linear combination $\mathbf{y}_c$ from the initial state $\mathbf{y}_0$ and target state $\mathbf{y}_f$. Then, our module decomposes the prediction of the clean sampled trajectory into linear and nonlinear modes, overcoming the limitations of single-network approaches that struggle to model both simultaneously. The theoretical foundation is as follows: $\mathbf{y}_c$ is the conditional input, and our denoiser is designed to output the clean state trajectory $\hat{\mathbf{x}}^0$, expressed as a vector function $\mathbf{f}(\mathbf{y}_c)$. It admits a vector Taylor expansion at $\mathbf{y}_c = \mathbf{0}$ as $\hat{\mathbf{x}}^0 = \mathbf{f}(\mathbf{y}_c) = \mathbf{C}_1\mathbf{y}_c + \mathbf{y}_c^T\mathbf{C}_2\mathbf{y}_c + \mathcal{O}(||\mathbf{y}_c||^3)$. For linear systems, only the first-order term remains. For nonlinear systems, by neglecting higher-order terms for simplicity, we can decompose the prediction into linear and nonlinear quadratic modes.

This decomposition imposes a strong inductive bias: modeling a dominant linear part and a subtle nonlinear correction is a more stable and sample-efficient task than forcing a monolithic network to learn the entire complex function from scratch. We implement this with a dual-UNet architecture, as illustrated in Figure 1. The network is conditioned on a vector $\mathbf{y}_c = f_c([\mathbf{y}_0, \mathbf{y}_f]) \in \mathbb{R}^{B \times C_1}$, which is generated from the start and target states via a bias-less linear layer $f_c$.

Given the noisy trajectory $\mathbf{x}^k \in \mathbb{R}^{B \times T \times N}$ and time embedding $\mathbf{k}_{\text{emb}}$, the first stage produces a linear prediction $\mathbf{O}_1 \in \mathbb{R}^{B \times T \times N}$ from first-order coefficients $\mathbf{C}_1$:

$$\mathbf{C}_1 = \text{UNet}_1(\mathbf{x}^k, \mathbf{k}_{\text{emb}}), \quad \mathbf{O}_1 = \text{reshape}(\mathbf{C}_1) \cdot \mathbf{y}_c. \tag{4}$$

In the second stage, $\text{UNet}_2$ combines the input $\mathbf{x}^k$ and intermediate features $\mathbf{C}_1$ to generate quadratic coefficients $\mathbf{C}_2$, which yield a nonlinear correction term $\mathbf{O}_2 \in \mathbb{R}^{B \times T \times N}$:

$$\mathbf{C}_2 = \text{UNet}_2([\mathbf{x}^k, \mathbf{C}_1], \mathbf{k}_{\text{emb}}), \quad \mathbf{O}_2 = \mathbf{y}_c^T \cdot \text{reshape}(\mathbf{C}_2) \cdot \mathbf{y}_c. \tag{5}$$

The final denoised prediction is the sum of these components:

$$\hat{\mathbf{x}}^0 = \mathbf{O}_1 + \mathbf{O}_2 \in \mathbb{R}^{B \times T \times N}. \tag{6}$$

Here, $\mathbf{C}_1 \in \mathbb{R}^{B \times T \times (N \times C_1)}$ and $\mathbf{C}_2 \in \mathbb{R}^{B \times T \times (C_1 \times N \times C_1)}$ represent the learned coefficient tensors.

### 4.3 Guided Self-finetuning (GSF)

Randomly generated training data cannot guarantee coverage of optimal scenarios. To generate near-optimal controls that may deviate significantly from the training distribution, we propose leveraging the model's initially generated data (under the guidance of the cost function), which naturally deviates from the training distribution toward optimality, for iterative retraining to systematically expand the exploration space. This approach maintains physical consistency by ensuring generated samples adhere to the underlying system dynamics.

As shown in Figure 1, our methodology involves extracting control sequences from the generated samples (i.e., the output of inverse dynamics $\tilde{\mathbf{u}}^0$) and reintroducing them into the system to interact and generate corresponding state sequences $\tilde{\mathbf{y}}^0$. Together, we add the renewed $[\tilde{\mathbf{u}}^0, \tilde{\mathbf{y}}^0]$ to the retrain data pool used for a new round of fine-tuning, notably without requiring explicit system parameter identification. We iterate this process over multiple rounds specified by a hyperparameter, systematically expanding the model's exploration space to progressively approach the optimal control policy. Denote the sampling process under cost $J$'s guidance and the following interacting process as $[\tilde{\mathbf{u}}^0, \tilde{\mathbf{y}}^0] = \mathcal{S}(\mathbf{x}^K, \mathbf{y}_0^*, \mathbf{y}_f, J, \Phi)$. The process can be formulated as:

$$[\tilde{\mathbf{u}}^0, \tilde{\mathbf{y}}^0] = \mathcal{S}_{(\mathbf{x}^K, \mathbf{y}_0^*, \mathbf{y}_f) \sim D}(\mathbf{x}^K, \mathbf{y}_0^*, \mathbf{y}_f, J, \Phi), \tag{7}$$

$$D = [D, [\tilde{\mathbf{u}}^0, \tilde{\mathbf{y}}^0]], \tag{8}$$

where $D$ is the training set and $\mathcal{S}(\mathbf{x}^K, \mathbf{y}_0^*, \mathbf{y}_f, J, \Phi)$ denotes the full process of generating a guided trajectory using cost $J$ and then interacting with the system dynamics $\Phi$ to get the corresponding state sequence. We provide the algorithm form of SEDC in Appendix A.

## 5 Experiments

### 5.1 Experiment Settings

We conducted experiments on three nonlinear systems, following the instructions in the previous works for data synthesis. These systems include: the 1-D Burgers dynamics (Hwang et al., 2022; Wei et al., 2024b), which is a high-order (128-dim) dynamic system for studying nonlinear wave propagation and turbulent fluid flow; the Kuramoto dynamics (Acebrón et al., 2005; Baggio et al., 2021; Gupta et al., 2022), which is essential for understanding synchronization phenomena in complex networks and coupled oscillator systems; the inverted pendulum dynamics (Boubaker, 2013), which represents a classical benchmark problem in nonlinear control theory and robotic systems; and the Jellyfish locomotion, a more challenging, high-dimensional 2D PDE control task for validating the scalability of our approach (Appendix F.2). For each system, we generated control/state trajectory data using the finite difference method and selected 50 trajectories as the test set. We assume full state observability throughout the experiments. Detailed descriptions of the system dynamics equations and data synthesis procedures are provided in Appendix B. Implementation details and time consumption reports are provided in Appendix C.

We evaluate two metrics which is crucial in complex system control: **Target Loss**, the mean-squared-error (MSE) of $\mathbf{y}_T$ and desired target $\mathbf{y}_f$, i.e. $\frac{1}{N}\|\mathbf{y}_T - \mathbf{y}_f\|^2$. (Note that $\mathbf{y}_T$ is obtained by simulating the real system using the control inputs generated by each method, along with the given initial state conditions, rather than extracted from the sample trajectories of the diffusion-based methods); **Energy** $J = \int_0^T |\mathbf{u}(t')|^2 dt'$, which measures the cumulative control effort required to achieve the target state. Lower values of both metrics indicate better performance.

**Baselines.** We select the following state-of-the-art(SOTA) baseline methods for comparison. For traditional control approaches, we employ the classical PID (Proportional-Integral-Derivative) controller (Li et al., 2006), which remains widely used in industrial applications. For supervised learning ,we employ Behavioral Cloning (BC) (Pomerleau, 1988), an established imitation learning approach. In terms of reinforcement learning methods, we incorporate BPPO (Zhuang et al., 2023), a state-of-the-art algorithm. For diffusion-based methods, we include several recent prominent approaches: DecisionDiffuser (DecisionDiff) (Ajay et al., 2022), which is a SOTA classifier-free diffusion-based planner; AdaptDiffuser (Liang et al., 2023), which enhances DecisionDiffuser with a self-tuning mechanism through interaction with the environment; RDM (Zhou et al., 2024), which adaptively

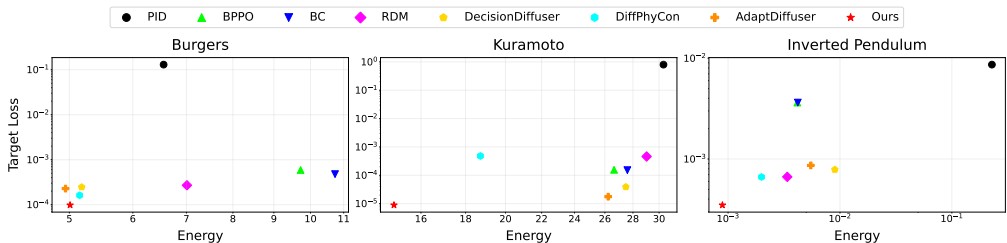

Figure 2: Comparison of target loss and energy cost $J$ across different datasets. The closer the data point is to the bottom left, the better the performance.

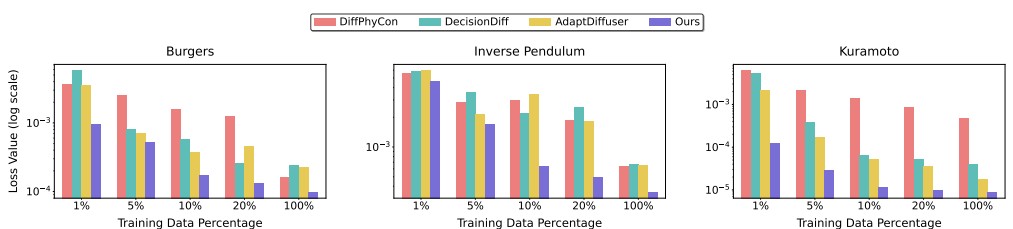

Figure 3: Sample-efficiency comparison on Burgers, Kuramoto and Inverse Pendulum dynamics.

determines the timing of complete control sequence sampling; and DiffPhyCon (Wei et al., 2024b), which is specifically designed for controlling complex physical systems. Detailed descriptions of the baselines are included in Appendix D.

## 5.2 OVERALL CONTROL PERFORMANCE

In Figure 2, we compare different methods' performance across three dynamical systems using two-dimensional coordinate plots, where proximity to the lower-left corner indicates better trade-offs between control accuracy and energy efficiency. Since unstable control can lead to system failure regardless of energy efficiency, we prioritize control accuracy and report metrics at each method's minimum Target Loss.

**SEDC consistently achieves state-of-the-art performance.** Our method secures the position closest to the origin in all three datasets, demonstrating the best balance between accuracy and efficiency. Specifically, SEDC achieves the lowest Target Loss across all systems, outperforming the strongest baselines by 39.5%, 49.4%, and 47.3% in the Burgers, Kuramoto, and IP systems, respectively. This highlights its superior capability in learning complex dynamics. In terms of energy cost, SEDC leads on the Kuramoto and IP systems and remains highly competitive on the Burgers system.

**Analysis of Baselines.** The results reveal clear performance tiers among different method families. Traditional PID control shows the poorest performance, as system complexity exacerbates the difficulties in PID control and tuning. RL-based methods are competitive against some diffusion- but sacrifice Target Loss performance and underperform compared to Diffusion-based methods in other systems. Diffusion-based methods demonstrate superior overall performance, as they better capture long-term dependencies in system dynamics compared to traditional and RL methods, avoiding myopic failure modes and facilitating global optimization of long-term dynamics.

Due to space constraints, Appendices E and G provide detailed numerical results, including standard errors, Pareto frontier comparisons, and control dynamics visualizations. A comparison with a learning-based MPC in Appendix F.1 further underscores the advantages of our global planning approach. While our main experiments assume system invertibility and full-state observability, we validate SEDC's applicability to non-invertible systems and its robustness to significant observation noise in Appendices F.4 and F.7. Furthermore, Appendices F.2 and F.3 validate the method's effectiveness on high-dimensional and complex real-world systems.

## 5.3 SAMPLE EFFICIENCY

To evaluate the sample efficiency of diffusion-based methods, we conducted experiments on all the systems using varying proportions of the full training dataset. Specifically, we trained models using

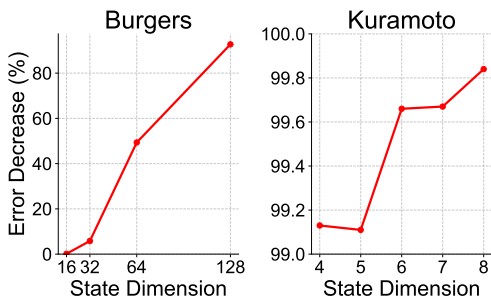
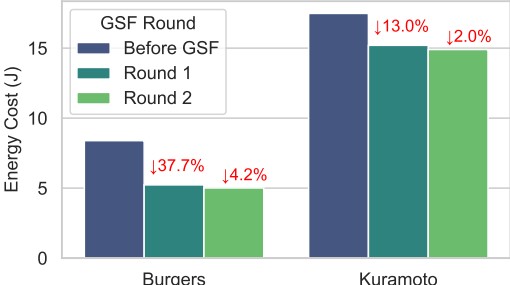

Figure 4: Target Loss decrease rate (reduction in target loss achieved by SEDC compared to *w/o DSD*) across different state dimensions in Burgers and Kuramoto.

Figure 5: Energy cost ($J$) reduction across GSF rounds. The first round yields substantial improvements, with subsequent rounds showing convergence.

1%, 5%, 10%, 20%, and 100% of the available data and assessed their performance using the Target Loss metric on a held-out test set. Figure 3 demonstrates our method's superior performance in controlling Burgers and Kuramoto systems compared to state-of-the-art baselines. In all systems, our approach achieves significantly lower target loss values across all training data percentages. Most notably, with only 10% of the training data, our method attains a target loss of 1.71e-4 for Burgers, 1.12e-5 for Kuramoto, and 6.35e-4 for Inverse Pendulum, matching(-5.5% in Burgers) or exceeding(+36.4% in Kuramoto and +1.2% in Inverse Pendulum) the performance of best baseline methods trained on the complete dataset. This indicates our method can achieve state-of-the-art performance while requiring only 10% of the training samples.

Among the baselines, DiffPhyCon's complex training requirement (Sec. 2) makes it particularly sample-inefficient. While AdaptDiffuser's self-tuning improves upon DecisionDiffuser, SEDC's advantage stems from its efficient dynamics learning and a filter-free finetuning strategy. Unlike AdaptDiffuser, our GSF mechanism integrates all guided trajectories without discriminator filtering, promoting data diversity and a better exploration-exploitation balance.

## 5.4 ABLATION STUDY

**Overall ablation study.** We explore the main performance against each ablation of the original SEDC. Specifically, *w/o DSD* removes the inverse dynamics, unifying the diffusion of system state and control input, i.e. $\mathbf{x} = [\mathbf{u}, \mathbf{y}]$. Therefore, the diffusion model is required to simultaneously capture the temporal information and implicit dynamics of the control and system trajectory. Note that the inpainting mechanism and gradient guidance are retained. *w/o DMD* removes the decomposition design, resulting in a single 1-D Unet structure

Table 1: Performance comparison of different ablations across multiple datasets. **Target loss** results with 10% and 100% training sample for each method are reported. The best, second-best and worst results of each row are highlighted in **bold**, underlined and *italics*, respectively.

| System | Ratio | Ours | w/o DSD | w/o DMD | w/o GSF |
|---|---|---|---|---|---|
| Burgers | 10% | **1.74e-4** | *1.00e-3* | 3.78e-4 | 6.67e-4 |
| | 100% | **9.80e-5** | *8.71e-4* | 2.28e-4 | 2.62e-4 |
| Kuramoto | 10% | **1.12e-5** | *4.15e-3* | 5.21e-5 | 4.77e-5 |
| | 100% | **8.90e-6** | *5.43e-3* | 1.76e-5 | 3.88e-5 |
| IP | 10% | **6.21e-4** | *1.58e-3* | 1.10e-3 | 2.00e-3 |
| | 100% | **3.49e-4** | *1.37e-3* | 6.64e-4 | 7.85e-4 |

as the denoising network, following DecisionDiff (Ajay et al., 2022). Finally, *w/o GSF* reports the performance without iterative self-finetuning, which means the model only uses the original dataset to train itself. To show the sample-efficiency performance, we also investigate the results under less amount of training sample(10%). For *w/o DMD* and *w/o DSD*, we adjust the number of trainable parameters at a comparable level against the original version.

Table 1 shows the Target Loss performance of different ablations of SEDC across multiple datasets and different training sample ratios. As can be seen, removing any component leads to a certain decrease in performance, whether the training data is limited or not, demonstrating the effectiveness of each design. The most significant performance drops are often observed in *w/o DSD*, highlighting the

importance of explicit learning of dynamics in complex systems. *w/o DMD* exhibits the lowest decline across the three systems. This is because the single-Unet-structured denoising network can already capture the nonlinearity to some extent, but not as good as the proposed decomposition approach. with 10% of training data, removing individual components still led to noticeable performance degradation, and the patterns consistent with the full dataset results. This demonstrates that our designs remain effective in low-data scenarios.

**Effectiveness of DSD.** To evaluate DSD's effectiveness against the curse of dimensionality, we compared the performance of original and *w/o DSD* models across Kuramoto systems with dimensions ranging from $N = 4$ to $N = 8$. Experimental results (Figure 4) show that performance degradation from *w/o DSD* increases proportionally with system dimensionality, demonstrating DSD's enhanced effectiveness in higher-dimensional systems and validating its capability to address dimensionality challenges. Further, we can observe that the decrease changes more rapidly with increasing dimensionality compared to Kuramoto, indicating that DSD exhibits heightened sensitivity to dimensionality in higher-dimensional systems. We also perform case study to investigate the effectiveness of dynamical learning in Appendix F.5.

**Effectiveness of DMD.** To investigate the contribution of DMD's dual-Unet architecture to nonlinearity learning, we conducted experiments on the Kuramoto system with varying degrees of nonlinearity (controlled by the coefficient $\gamma \in \{1, 2, 4\}$ of the nonlinear sinusoidal term, where larger values indicate stronger nonlinearity). We compared the performance between using only the linear intermediate output ($\hat{\mathbf{x}}_0 = \mathbf{O}_1$) of the denoising network and the original nonlinear

Table 2: Performance degradation using different denoiser output with varying nonlinearity strength $\gamma$ in the Kuramoto system.

| $\gamma$ | 1 | 2 | 4 |
|---|---|---|---|
| $\mathbf{O}_1 + \mathbf{O}_2$ | 8.90e-6 | 2.78e-5 | 3.89e-5 |
| $\mathbf{O}_1$ | 1.42e-5 | 4.73e-5 | 8.52e-5 |
| Dec. (%) | 37.3 | 41.2 | 54.3 |

output ($\hat{\mathbf{x}}_0 = \mathbf{O}_1 + \mathbf{O}_2$) in terms of Target Loss. The Dec. indicates the reduction in target loss achieved by nonlinear output $\mathbf{O}_1 + \mathbf{O}_2$ compared to linear output $\mathbf{O}_1$. As shown in Table 2, the magnitude of loss reduction increases proportionally with the nonlinearity strength $\gamma$, indicating that the quadratic term exhibits enhanced capability in capturing nonlinear dynamics as the system's nonlinearity intensifies. This demonstrates both the significance of the nonlinear branch $\mathbf{O}_2$ in capturing strong nonlinear dynamics and the effectiveness of decoupling linear and nonlinear modes in handling system nonlinearity. We also show that DMD is sufficient for modeling even higher-order dynamics in Appendix F.6.

**Effectiveness of GSF.** To validate GSF's ability to refine the control policy towards lower energy cost, we tracked test-set energy consumption across finetuning rounds. As shown in Figure 5, the first GSF round yields a dramatic improvement over the initial model, reducing energy by 37.7% (Burgers) and 13.0% (Kuramoto). Subsequent rounds offer minor refinements (4.2% and 2.0% respectively), indicating convergence to a near-optimal policy. This confirms GSF effectively guides the model beyond its initial suboptimal training data to discover more energy-efficient control solutions.

# 6 CONCLUSION

In this paper, we presented SEDC, a novel sample-efficient diffusion-based framework for complex nonlinear system control. By synergistically integrating Decoupled State Diffusion (DSD), Dual-Mode Decomposition (DMD), and Guided Self-finetuning (GSF), SEDC achieves superior control performance with remarkable data efficiency. Our experiments show that SEDC can match state-of-the-art accuracy using just 10% of the training data. The framework's robustness is further validated through rigorous testing on high-dimensional PDEs, non-invertible dynamics, and noisy observations, confirming the broad applicability and effectiveness of our design principles. These results mark a significant advancement in developing practical and sample-efficient solutions for complex system control.

ETHICS STATEMENT

The research presented in this paper focuses on foundational methodologies for the control of simulated complex nonlinear systems, such as those governing fluid dynamics and oscillator networks. The work does not involve human subjects, personally identifiable information, or sensitive data, thus posing no direct ethical concerns regarding privacy or data security. The datasets used for training and evaluation were synthetically generated based on established physical models from prior literature. While the developed control techniques could potentially be applied to real-world systems in the future, most of this work is confined to a theoretical and computational scope. We have adhered to the ICLR Code of Ethics throughout this research and foresee no negative societal impacts stemming directly from the publication of these results.

REPRODUCIBILITY STATEMENT

The source code for our proposed method, SEDC, along with the scripts used for data generation, model training, and evaluation for all experiments, is provided in the abstract as an anonymous downloadable link. Detailed descriptions of the system dynamics, governing equations, and data synthesis procedures for each benchmark are provided in Appendix B. Hyperparameter settings, model architecture specifics, and other implementation details are documented in Appendix C. The appendix also contains additional ablation studies and supplementary experiments (Appendices F.1-F.7) that further validate our claims and provide a comprehensive basis for reproduction.

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

# A  ALGORITHM FORM OF SEDC

---

**Algorithm 1:** SEDC: Training and finetuning

---

**Input:** Initial dataset $\mathcal{D}_0$, diffusion steps $K$, guidance strength $\lambda$, self-finetuning rounds $R$,
  forward dynamics $f_{\text{forward}}$
**Output:** Optimized trajectory $\mathbf{y}_{0:T}^0$, controls $\mathbf{u}_{0:T}^0$
**Function** *Initial Training($\mathcal{D}_0$)*
  **while** *not converged* **do**
    Sample batch $(\mathbf{y}_{0:T}, \mathbf{u}_{0:T}) \sim \mathcal{D}_0$
    Sample $k \sim \mathcal{U}\{1, ..., K\}$, $\epsilon \sim \mathcal{N}(0, I)$
    Corrupt states: $\mathbf{y}^k = \sqrt{\bar{\alpha}^k}\mathbf{y} + \sqrt{1 - \bar{\alpha}^k}\epsilon$
    Predict clean states: $\hat{\mathbf{y}}^k = \mathcal{G}_\theta(\mathbf{y}^k, k, \mathbf{y}_0^*, \mathbf{y}_f)$
    Predict controls: $\tilde{\mathbf{u}}_t = f_\phi(\mathbf{y}_t, \mathbf{y}_{t+1})$
    Compute losses: $\quad L_{\text{diff}} = \|\mathbf{y} - \hat{\mathbf{y}}^k\|^2$
      $L_{\text{inv}} = \|\mathbf{u}_t - \hat{\mathbf{u}}_t\|^2$
    Update $\theta, \phi$ with $\nabla(L_{\text{diff}} + L_{\text{inv}})$

**for** $r = 1$ *to* $R$ **do**
  **Guided Data Generation:**
  Initialize $\mathbf{y}^K \sim \mathcal{N}(0, I)$, sample $(\mathbf{y}_0^*, \mathbf{y}_f) \sim \mathcal{D}_{r-1}$
  **for** $k = K$ *downto* 1 **do**
    Predict $\hat{\mathbf{y}}^k = \mathcal{G}_\theta(\mathbf{y}^k, k, \mathbf{y}_0^*, \mathbf{y}_f)$
    Compute gradient: $g = \nabla_{\mathbf{y}^k} J(\tilde{\mathbf{y}}^k, \hat{\mathbf{u}}^k)$, where $\tilde{\mathbf{u}}_t^k = f_\phi(\hat{\mathbf{y}}_t^k, \hat{\mathbf{y}}_{t+1}^k)$
    Adjust mean: $\mu_\theta = \mu_\theta^{(\text{base})} - \lambda\Sigma^k g$
    Sample $\mathbf{y}^{k-1} \sim \mathcal{N}(\mu_\theta, \Sigma^k I)$
    Enforce constraints: $\mathbf{y}_0^{k-1} \leftarrow \mathbf{y}_0^*$, $\mathbf{y}_T^{k-1} \leftarrow \mathbf{y}_f$
  Recover controls: $\tilde{\mathbf{u}}_t^0 = f_\phi(\mathbf{y}_t^0, \mathbf{y}_{t+1}^0)$
  **System Interaction:**
  Generate $\tilde{\mathbf{y}}_{[0:T]}^0 = f_{\text{forward}}(\tilde{\mathbf{u}}_{[0:T]}^0, \mathbf{y}_0^*)$
  Augment dataset: $\mathcal{D}_r = \mathcal{D}_{r-1} \cup \{(\tilde{\mathbf{y}}_{[0:T]}^0, \tilde{\mathbf{u}}_{[0:T]}^0)\}$
  **Adaptive Fine-tuning:**
  **while** *validation loss decreases* **do**
    Sample batch from $\mathcal{D}_r$
    Perform training steps as in Initial Training

**return** Optimized $\theta, \phi$
*(Test process follows guided data generation with test conditions $(\mathbf{y}_0^*, \mathbf{y}_f)$ provided.)*

---

# B  DETAILED SYSTEM AND DATASET DESCRIPTION

## B.1  OVERALL INTRODUCTION

Our benchmark selection (Inverted Pendulum, Kuramoto, and Burgers) follows established control systems research practice, chosen for real-world relevance and diverse nonlinearity and complexity:

- Inverted Pendulum: state 2, control 1, timestep 128

- Kuramoto: state 8, control 8, timestep 15

- Burgers: state 128, control 128, timestep 10

## B.2  BURGERS DYNAMICS

The Burgers' equation is a governing law occurring in various physical systems. We consider the 1D Burgers' equation with the Dirichlet boundary condition and external control input $\mathbf{u}(t, x)$:

$$
\begin{cases}
\frac{\partial y}{\partial t} = -y \cdot \frac{\partial y}{\partial x} + \nu \frac{\partial^2 y}{\partial x^2} + \mathbf{u}(t, x) & \text{in } [0, T] \times \Omega \\
y(t, x) = 0 & \text{on } [0, T] \times \partial\Omega \\
y(0, x) = y_0(x) & \text{in } \{t = 0\} \times \Omega
\end{cases}
$$

Here $\nu$ is the viscosity parameter, and $y_0(\mathbf{x})$ is the initial condition. Subject to these equations, given a target state $y_d(x)$, the objective of control is to minimize the control error $\mathcal{J}_{\text{actual}}$ between $y_T$ and $y_d$, while constraining the energy cost $\mathcal{J}_{\text{energy}}$ of the control sequence $\mathbf{u}(t, x)$.

We follow instructions in Wei et al. (2024b) to generate a 1D Burgers' equation dataset. Specifically, for numerical simulation, we discretized the spatial domain [0,1] and temporal domain [0,1] using the finite difference method (FDM). The spatial grid consisted of 128 points, while the temporal domain was divided into 10000 timesteps. We initiated the system with randomly sampled initial conditions and control inputs drawn from specified probability distributions. This setup allowed us to generate 90000 trajectories for training and 50 trajectories for testing purposes.

### B.3 KURAMOTO DYNAMICS

The Kuramoto model is a paradigmatic system for studying synchronization phenomena. We considered a ring network of $N = 8$ Kuramoto oscillators. The dynamics of the phases (states) of oscillators are expressed by:

$$
\dot{\theta}_{i,t} = \omega + \gamma(\sin(\theta_{i-1,t-1} - \theta_{i,t-1}) + \sin(\theta_{i+1,t-1} - \theta_{i,t-1})) + u_{i,t-1}, \quad i = 1, 2, ..., N. \quad (9)
$$

For the Kuramoto model, we generated 20,000 samples for training and 50 samples for testing. The initial phases were sampled from a Gaussian distribution $\mathcal{N}(0, I)$, and the random intervention control signals were sampled from $\mathcal{N}(0, 2I)$. The system was simulated for $T = 16$ time steps with $\omega = 0$, following Baggio et al. (2021). The resulting phase observations and control signals were used as the training and test datasets.

### B.4 INVERTED PENDULUM DYNAMICS

The inverted pendulum is a classic nonlinear control system. The dynamics can be represented by:

$$
\frac{d^2\theta}{dt^2} = \frac{g}{L}\sin(\theta) - \frac{\mu}{L}\frac{d\theta}{dt} + \frac{1}{mL^2}u
$$

where $\theta$ is the angle from the upward position, and $u$ is the control input torque. The system parameters are set as: gravity $g = 9.81$ m/s², pendulum length $L = 1.0$ m, mass $m = 1.0$ kg, and friction coefficient $\mu = 0.1$.

To generate the training dataset, we simulate 90,000 trajectories for training and 50 for testing with 128 time steps each, using a time step of 0.01s. For each trajectory, we randomly sample initial states near the unstable equilibrium point with $\theta_0 \sim \mathcal{U}(-1, 1)$ and $\dot{\theta}_0 \sim \mathcal{U}(-1, 1)$, and generate control inputs from $u \sim \mathcal{U}(-0.5, 0.5)$. The resulting dataset contains the state trajectories and their corresponding control sequences.

## C IMPLEMENTATION DETAILS

### C.1 IMPLEMENTATION OF SEDC

In this section, we describe various architectural and hyperparameter details:

- The temporal U-Net (1D-Unet) (Janner et al., 2022) in the denoising network consists of a U-Net structure with 4 repeated residual blocks. Each block comprises two temporal convolutions, followed by group normalization, and a final Mish nonlinearity. The channel dimensions of the downsample layers are $1, 2, 4 * statedimension$. Timestep embedding is

produced by a Sinusoidal Positional Encoder, following a 2-layer MLP, and the dimension of this embedding is 32. The dimension of condition embedding is the same as the system state dimension.

- We represent the inverse dynamics $f_\phi$ with an autoregressive model with 64 hidden units and ReLU activations. The model autoregressively generates control outputs along the control dimensions.

- We train $\mathbf{x}_\theta$ and $f_\phi$ using the Adam optimizer with learning rates from {1e-3, 5e-3, 1e-4}. The exact choice varies by task. Moreover, we also use a learning rate scheduler with step factor=0.1. Training batch size is 32.

- We use $K = 128$ diffusion steps.

- We use a guidance scale $\lambda \in \{0.01, 0.001, 0.1\}$ but the exact choice varies by task.

## C.2 TRAINING AND INFERENCE TIME ANALYSIS

Table 3: Approximate Training Time Comparison of Different Models on Various Datasets (in hours)

| Dataset/System | DecisionDiffuser | RDM | DiffPhyCon | AdaptDiffuser | SEDC |
|---|---|---|---|---|---|
| Burgers | 2.5 | 2.5 | 3.0 | 2.5 | 2.5 |
| Kuramoto | 1.5 | 1.5 | 1.5 | 1.0 | 1.0 |
| IP | 1.0 | 1.0 | 1.5 | 1.0 | 0.5 |

Table 4: Approximate Inference Time Comparison of Different Models on Various Datasets (in seconds)

| Dataset/System | DecisionDiffuser | RDM | DiffPhyCon | AdaptDiffuser | SEDC |
|---|---|---|---|---|---|
| Burgers | 3.0 | 4.0 | 6.0 | 4.0 | 4.0 |
| Kuramoto | 1.0 | 1.5 | 2.0 | 1.5 | 1.5 |
| IP | 0.5 | 1.0 | 1.0 | 0.5 | 0.5 |

The diffusion-based methods are trained on single NVIDIA GeForce RTX 4090 GPU. We evaluate the training and inference time of all the diffusion-based methods evaluated in the experiment session. As shown in Table 3, we compare the training efficiency of different models across various datasets. DiffPhyCon consistently shows longer training times compared to other methods, because it requires training two models that learn the joint distribution and the prior distribution respectively, increasing its training time consumption. The training times of DecisionDiffuser, RDM, and AdaptDiffuser are generally comparable, while SEDC demonstrates relatively efficient training performance across most datasets. This may be because of the proposed designs that not only improve sample efficiency but also improve learning efficiency.

The inference time comparison in Table 4 reveals that DiffPhyCon requires longer execution time compared to other models, because it needs to sample from two learned distributions in the denoising process. RDM achieves relatively slower inference speeds than DecisionDiffuser, AdaptDiffuser, and SEDC, because RDM replans during inference, increasing planning time. Notably, all models exhibit shorter training and inference times on the IP dataset, suggesting the influence of system complexity on computational efficiency.

## D  BASELINES DESCRIPTION

### D.1  PID

PID (Proportional-Integral-Derivative) control is a classical feedback control methodology that has been widely adopted in industrial applications. The control signal is generated by computing the weighted sum of proportional, integral, and derivative terms of the error. The control law can be expressed as:

$$u(t) = K_p e(t) + K_i \int_0^t e(\tau)d\tau + K_d \frac{d}{dt}e(t)$$

While PID controllers exhibit robust performance and require minimal system modeling, their effectiveness may be compromised when dealing with highly nonlinear or time-varying systems, necessitating frequent parameter tuning.

### D.2 BC, BPPO

Behavior Cloning (BC) represents a supervised imitation learning paradigm that aims to learn a direct mapping from states to actions by minimizing the deviation between predicted actions and expert demonstrations. Despite its implementation simplicity and sample efficiency, BC suffers from distributional shift, where performance degradation occurs when encountering states outside the training distribution. The objective function can be formulated as:

$$L_{BC}(\theta) = \mathbb{E}_{(s,a)\sim\mathcal{D}}[-\log \pi_\theta(a|s)]$$

where $\mathcal{D}$ denotes the expert demonstration dataset.

Behavior-guided PPO (BPPO) presents a hybrid approach that integrates behavior cloning with Proximal Policy Optimization. By incorporating a behavioral cloning loss term into the PPO objective, BPPO facilitates more efficient policy learning while maintaining the exploration capabilities inherent to PPO. The composite objective function is defined as:

$$L_{BPPO}(\theta) = L_{PPO}(\theta) + \alpha L_{BC}(\theta)$$

where $\alpha$ serves as a balancing coefficient between the PPO and BC objectives.

Each method exhibits distinct characteristics: BC demonstrates effectiveness when abundant high-quality expert demonstrations are available. BPPO leverages the synergy between expert knowledge and reinforcement learning for complex control scenarios.

### D.3 DIFFUSION-BASED METHODS

- **DecisionDiffuser:**
  A novel approach that reformulates sequential decision-making as a conditional generative modeling problem rather than a reinforcement learning task. The core methodology involves modeling policies as return-conditional diffusion models, enabling direct learning from offline data without dynamic programming. The model can be conditioned on various factors including constraints and skills during training.

- **DiffPhyCon:**
  A diffusion-based method for controlling physical systems that operates by jointly optimizing a learned generative energy function and predefined control objectives across entire trajectories. The approach incorporates a prior reweighting mechanism to enable exploration beyond the training distribution, allowing the discovery of diverse control sequences while respecting system dynamics.

- **AdaptDiffuser:**
  An evolutionary planning framework that enhances diffusion models through self-evolution. The method generates synthetic expert data using reward gradient guidance for goal-conditioned tasks, and employs a discriminator-based selection mechanism to identify high-quality data for model fine-tuning. This approach enables adaptation to both seen and unseen tasks through continuous model improvement.

- **RDM:**
  A replanning framework for diffusion-based planning systems that determines replanning timing based on the diffusion model's likelihood estimates of existing plans. The method introduces a mechanism to replan existing trajectories while maintaining consistency with original goal states, enabling efficient bootstrapping from previously generated plans while adapting to dynamic environments.

# E  DETAILED RESULTS OF FIGURE 2

Table 5: **Performance comparison of different models across three datasets.** TL (Target Loss) and $J$ (Energy) are reported, with lower values indicating better performance for both metrics. We report the mean and the standard error over 5 random seeds. Following previous work(e.g. Ajay et al. (2022)), we highlight the best-performed results in **bold**.

| Model | Burgers | | Kuramoto | | IP | |
|---|---|---|---|---|---|---|
| | TL | $J$ | TL | $J$ | TL | $J$ |
| PID | 1.30e-1 | 6.56 | 7.99e-1 | 30.35 | 8.64e-3 | 2.28e-1 |
| BPPO | 5.90e-4 | 9.72 | 1.56e-4 | 26.64 | 3.63e-3 | 4.16e-3 |
| BC | 4.78e-4 | 10.73 | 1.52e-4 | 27.59 | 3.63e-3 | 4.20e-3 |
| DecisionDiff | 2.46e-4 | 5.18 | 3.88e-5 | 27.48 | **6.65e-4** | **9.00e-4** |
| RDM | 2.70e-4 | 7.01 | 4.60e-4 | 29.03 | 7.85e-4 | 3.38e-3 |
| DiffPhyCon | **1.62e-4** | **5.15** | 4.80e-4 | 18.72 | 6.63e-4 | 1.99e-3 |
| AdaptDiffuser | 2.28e-4 | **4.65** | **1.76e-5** | 26.23 | 8.64e-4 | 5.49e-3 |
| Ours | **9.80±5.6e-5** | **5.01±0.6** | **8.90±3.1e-6** | **14.90±0.8** | **3.49±2.6e-4** | **8.90±0.9e-4** |

We leverage 2-D plots in the main paper to better illustrate the performance comparison of all the methods. Here we provide the provides the corresponding numerical results in detail in Table 5 and pareto frontiers in Fig 6. Results confirm our method maintains competitive performance across all datasets.

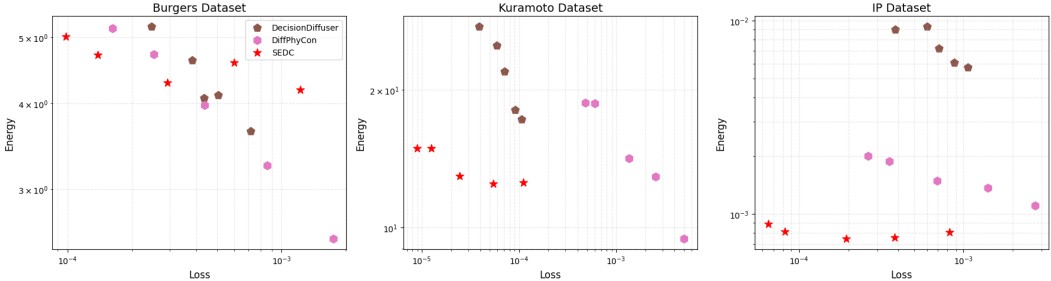

Figure 6: Pareto frontier of target loss and energy cost $J$ across different datasets of our method and two SOTA baselines DecisionDiffuser and DiffPhyCon. The closer the data point is to the bottom left, the better the performance.

# F  SUPPLEMENTARY EXPERIMENTS

## F.1  COMPARISON AGAINST MPC

Table 6: Comparison of SEDC (Ours) and Model Predictive Control (MPC) across three control tasks. Results show target loss (MSE), control cost (J), and inference time in seconds. We implemented the MPC that uses a neural network architecture with residual connections to learn system dynamics from data, then solves finite-horizon optimization problems at each timestep using the gradient of the summation of target loss and control energy, which iteratively refines the sampling distribution toward optimal control sequences. The control horizons of Kuramoto,Burgers and IP are 15, 10 and 128.

| Model | Kuramoto | | | Burgers | | | IP | | |
|---|---|---|---|---|---|---|---|---|---|
| | Target loss | J | Time(s) | Target loss | J | Time(s) | Target loss | J | Time(s) |
| MPC | 9.10e-03 | 0.34 | ~50 | 1.89e-01 | 0.001 | ~30 | 4.33e-04 | 1.01e-03 | >1000 |
| Ours | 8.90e-06 | 14.9 | ~1.5 | 9.80e-05 | 5.01 | ~2.5 | 3.49e-04 | 8.90e-04 | ~0.5 |

We additionally compare SEDC with learning-based Model Predictive Control (MPC). We implement MPC in a data-driven way, where we train an MLP for the forward model. Results in Table 6 show MPC achieves higher target losses across all tasks, likely due to error accumulation. MPC also has significantly longer inference times (e.g. ¿1000 vs. 0.5) that increase with control horizon. SEDC directly maps initial/target states to complete control trajectories, avoiding compounding errors and reducing computation time.

### F.2 VALIDATION ON HIGH-DIMENSIONAL 2D PDE CONTROL: JELLYFISH LOCOMOTION

To address the critical question of our method's scalability and effectiveness on truly high-dimensional control problems, we conducted additional experiments on a challenging 2D PDE control benchmark: the locomotion of a jellyfish. This task represents a state-of-the-art challenge in data-driven control of physical systems, involving complex fluid-solid interactions governed by the 2D incompressible Navier-Stokes equations (Wei et al., 2024b). Unlike the systems in the main text, this benchmark provides a testbed with a significantly higher state-space dimension, allowing for a rigorous evaluation of SEDC's scalability.

**Experimental Setup** The objective is to control the opening angle of the jellyfish's wings to achieve a target locomotion pattern. The system state is a high-dimensional PDE field representing the fluid velocity and pressure, while the control input is a scalar time series representing the wing angle.

- **State Representation:** Each state at a given timestep is represented by a $3 \times 32 \times 32$ tensor, resulting in a **state dimension of 3,072**. This is a substantial increase in complexity compared to the 1D systems.
- **Dataset:** We generated a dataset of 20,000 control trajectories for training, following the standard procedure for this benchmark.

**Overall Performance Comparison** We first evaluated SEDC against strong diffusion-based baselines using the full training dataset. The results, shown in Table 7, assess control accuracy (Target Loss), energy efficiency (Energy), and computational cost (Training and Inference Time).

Table 7: Performance comparison on the 2D Jellyfish Locomotion control task (100% training data). SEDC achieves the best control accuracy with a competitive computational profile.

| Model | Target Loss ($\downarrow$) | Energy ($\downarrow$) | Train Time (hrs) | Inference Time (s) |
|---|---|---|---|---|
| DecisionDiffuser | 1.74e-4 $\pm$ 7.4e-5 | 2.016 $\pm$ 0.08 | 3.0 | 6.0 |
| AdaptDiffuser | 1.77e-4 $\pm$ 1.7e-5 | **2.001 $\pm$ 0.71** | 3.0 | 6.0 |
| DiffPhyCon | 1.77e-4 $\pm$ 5.4e-5 | 2.157 $\pm$ 0.21 | 4.0 | 8.0 |
| **SEDC (Ours)** | **1.70e-4 $\pm$ 1.3e-5** | 2.015 $\pm$ 0.43 | 3.0 | 6.5 |

The results demonstrate that SEDC achieves the best control accuracy (lowest Target Loss) among all methods, confirming that its architectural advantages for sample-efficient learning translate effectively to high-dimensional PDE systems. Furthermore, its computational cost remains on par with the fastest baselines, highlighting its practicality.

**Sample Efficiency Analysis** A key claim of our work is superior sample efficiency. To specifically validate this on a high-dimensional task, we compared the performance of SEDC against the strongest baseline (DecisionDiffuser) when trained on only 20% of the available data versus the full dataset.

Table 8: Sample efficiency comparison on the Jellyfish task. Results show the final Target Loss. SEDC trained on only 20% of the data outperforms the baseline trained on 100% of the data.

| Model | 100% Training Data | 20% Training Data |
|---|---|---|
| DecisionDiffuser | 1.74e-4 | 4.52e-4 |
| **SEDC (Ours)** | **1.70e-4** | **1.75e-4** |

As shown in Table 8, SEDC exhibits remarkable sample efficiency. When trained on just 20% of the data, its performance is statistically on par with the baseline trained on the full dataset (1.75e-4

vs 1.74e-4). In contrast, the baseline's performance degrades significantly when data is limited. This comprehensive validation on a high-dimensional PDE benchmark strongly supports our paper's central claim: SEDC provides a scalable and highly sample-efficient framework for the control of complex nonlinear systems.

## F.3   PERFORMANCE ON SWING DYNAMICS

Table 9: Performance comparison on the power grid swing dynamics system. This trajectories are sampled with 18 state variables, 9 control inputs, and 32 timesteps to model electrical grid behavior with high fidelity, incorporating realistic physical disturbances. The Target Loss (MSE between the final controlled states and the target zero states) and the control Energy are reported.

| Model | Target Loss | Energy |
|---|---|---|
| DecisionDiffuser | 0.0413 | 0.306 |
| RDM | 0.0609 | **0.237** |
| DiffPhyCon | 0.0479 | **0.236** |
| **Ours** | **0.0163**±0.011 | **0.229**±0.05 |

Since Kuramoto may oversimplify real-world complexity, we conduct additional experiments on swing dynamics (Baggio et al., 2021),which models real-world power grid behavior with higher fidelity and complexity. The background is about a line fault in the New England powergrid network comprising 39 nodes (29 load nodes and 10 generator nodes). Following the experiment design in Baggio et al. (2021), we use our model to compute an optimal point-to-point control from data to recover the correct operation of the grid. The goal is to operate the un-synchronized state of the generators to recover the steady states of zeros. The result is shown in Table 9, showing our method achieves lowest target loss, outperforming DecisionDiffuser by 60%, confirming that our approach's benefits extend to practical complex scenarios.

## F.4   SYSTEM APPLICABILITY AND THE INVERTIBILITY ASSUMPTION

Here, we clarify that our framework does not require mathematical invertibility and empirically validate its robustness on challenging non-invertible systems. Our model $f_\phi$ is a learned function approximator, not a formal analytical inverse. By minimizing the Mean Squared Error (MSE) during training, it learns the conditional expectation of the control given a state transition, $\mathbb{E}[\mathbf{u}|\mathbf{y}_t, \mathbf{y}_{t+1}]$.

- For an invertible system, this learned expectation converges to the unique correct control action.
- For a non-invertible system where multiple controls $\mathbf{u}_i$ could produce the same transition, the model learns a consistent and effective policy. For instance, in affine-in-control systems ($\dot{\mathbf{y}} = f(\mathbf{y}) + g(\mathbf{y})\mathbf{u}$), the linearity in the control term ensures that the learned expected control $\mathbb{E}[\mathbf{u}]$ still produces a valid transition.

The diffusion model first generates a feasible state trajectory, and the role of $f_\phi$ is simply to provide a suitable control sequence to realize it. This decoupled strategy of learning a plausible mapping is substantially more effective than attempting to model the complex, potentially multi-modal joint distribution of states and controls, as demonstrated below.

We conducted new experiments on two distinct non-invertible systems, comparing our full SEDC model against an ablation variant ('w/o DSD') that jointly diffuses states and controls.

**Experiment 1: Non-Affine, Non-Invertible MIMO System.**

We designed a 2-state $(x_1, x_2)$, 2-input $(u_1, u_2)$ system with a non-affine and non-invertible control term:

$$\dot{x}_1 = -x_1 + x_2$$
$$\dot{x}_2 = \sin(x_1) - 0.5x_2 + u_1^2 - u_2^2$$

The control term $u_1^2 - u_2^2$ makes it impossible to uniquely determine $u_1$ and $u_2$ from the states. As shown in Table 10, SEDC is significantly more accurate and energy-efficient.

Table 10: Performance on the non-affine, non-invertible MIMO system.

| Method | Target Loss | Energy |
|---|---|---|
| SEDC (Ours) | **5.0e-2** ± **1.2e-3** | **1.8e-2** ± **4.9e-4** |
| w/o DSD | 8.1e-2 ± 1.5e-3 | 0.60 ± 0.11 |

**Experiment 2: Non-Invertible, Rank-Deficient Linear System.**

Table 11: Performance on the rank-deficient linear system.

| Method | Target Loss | Energy |
|---|---|---|
| SEDC (Ours) | **5.4e-3** ± **3.1e-3** | **2.55** ± **1.14** |
| w/o DSD | 1.9e-2 ± 1.7e-3 | 3.66 ± 1.02 |

We also tested a linear system with a rank-deficient control matrix, where an infinite number of control inputs can yield the same effect. This represents a simpler class of non-invertibility. The results in Table 11 again show that SEDC's decoupled approach of learning an effective "average" control policy is a more robust strategy.

These experiments confirm that our DSD framework is robust and highly effective for a broad class of systems beyond those that are strictly invertible. Learning a plausible control mapping is a more sample-efficient and accurate strategy than modeling a complex joint state-action distribution.

### F.5 THE EFFECTIVENESS OF DYNAMICAL LEARNING

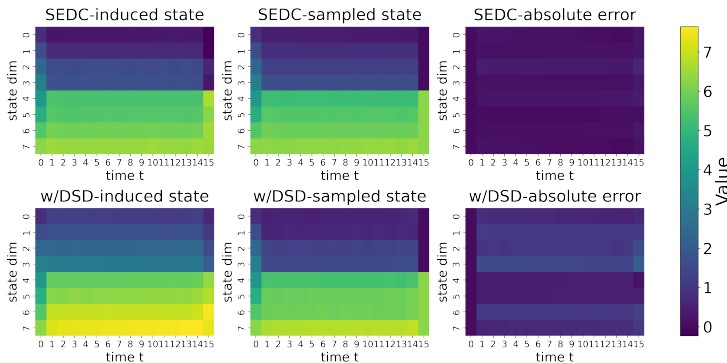

Figure 7: Comparison of State Trajectory Consistency between SEDC and *w/o DSD* Models. The heatmaps show induced states (left), sampled states (middle), and their absolute differences (right) for both SEDC (top) and *w/o DSD* (bottom) approaches under identical start-target conditions.

To investigate the effectiveness of dynamical learning, we compared the consistency between action sequences and diffusion-sampled state trajectories in models with and without DSD. While both approaches can sample state trajectories from diffusion samples, they differ in action generation: SEDC uses inverse dynamics prediction, whereas *w/o DSD* obtains actions directly from diffusion samples by simultaneously diffusing states and control inputs. We test both models using identical start-target conditions and visualize the state induced from the generated actions and the state sampled from the diffusion model, along with the difference (error) between the above two states in Figure 7. We can observe that SEDC's action-induced state trajectories showed significantly higher consistency with sampled trajectories compared to *w/o DSD*, demonstrating that DSD using inverse dynamics achieves more accurate learning of control-state dynamical relationships.

F.6 SUFFICIENCY OF THE 2ND-ORDER DMD ARCHITECTURE

To address whether our 2nd-order DMD approximation is sufficient for systems with richer nonlinearities, we conducted a targeted experiment on synthetic 2D systems with controlled nonlinear terms. We designed three systems: one purely linear, one with a quadratic term, and one with a cubic term. We compared our full DMD model against a single-UNet baseline and a linear-only ablation of our model.

Table 12: Performance (Target Loss) on synthetic systems with varying orders of nonlinearity. Our 2nd-order DMD model is sufficient to effectively control the 3rd-order system.

| System Dynamics | Single-UNet | DMD (Ours) | Linear-Only |
|---|---|---|---|
| 1st-Order (Linear) | 1.05e-6 | **8.68e-7** | 1.54e-6 |
| 2nd-Order (Quadratic) | 6.63e-6 | **5.20e-6** | 2.50e-4 |
| 3rd-Order (Cubic) | 7.00e-5 | **6.43e-5** | 5.80e-3 |

As shown in Table 12, our DMD model achieves the lowest error on the 3rd-order system, demonstrating its sufficiency. The performance of the Linear-Only model is nearly 90x worse, confirming that the nonlinear branch ($O_2$) is not redundant and is critical for capturing the system's dynamics. This principled, sample-efficient design proves robust even for complex systems beyond its explicit Taylor-expansion motivation.

F.7 ROBUSTNESS TO OBSERVATION NOISE

To evaluate the robustness of our method, a critical factor for real-world applicability, we conducted new experiments to validate SEDC's performance when trained on data corrupted by observation noise. This setup simulates practical scenarios where state measurements are imperfect.

**Experimental Setup** We added zero-mean Gaussian noise with varying standard deviations ($\sigma$) to the state observations in the training data for both the Kuramoto and Burgers systems. We then retrained our model and the strongest baselines from scratch on this noisy data and evaluated their control accuracy on a clean, noise-free test set.

**Results** The results, summarized in Table 13, show that SEDC consistently achieves superior control accuracy across all noise levels.

Table 13: Performance (Target Loss) comparison on noisy training data. Energy cost is shown in parentheses. SEDC demonstrates consistently higher accuracy under noisy conditions.

| System | Noise ($\sigma$) | SEDC (Ours) | AdaptDiffuser | DiffPhyCon |
|---|---|---|---|---|
| Kuramoto | 0 | **8.90e-6** (14.90) | 1.76e-5 (26.23) | 4.80e-4 (18.72) |
| | 0.001 | **1.21e-5** (17.19) | 8.00e-4 (16.24) | 3.95e-3 (11.50) |
| | 0.01 | **5.61e-4** (10.40) | 1.75e-3 (3.37) | 3.24e-3 (17.58) |
| | 0.1 | **3.14e-3** (0.42) | 3.96e-3 (0.98) | **3.14e-3** (2.22) |
| Burgers | 0 | **9.80e-5** (5.01) | 2.28e-4 (4.65) | 1.62e-4 (5.15) |
| | 0.001 | **1.86e-4** (5.01) | 2.81e-4 (4.73) | 6.05e-4 (5.11) |
| | 0.01 | **2.25e-4** (4.40) | 3.98e-4 (4.21) | 8.69e-4 (4.58) |
| | 0.1 | **6.97e-4** (4.25) | 1.22e-3 (3.75) | 2.89e-3 (3.13) |

Notably, on the Burgers system with medium noise ($\sigma = 0.01$), SEDC's target loss of 2.25e-4 is 1.8x better than AdaptDiffuser's and 3.9x better than DiffPhyCon's. This superior robustness stems from our key architectural innovations. DMD's decomposition helps capture the core system dynamics resiliently, while DSD's focus on learning a smoother state-only distribution prevents overfitting to noise, a common issue when modeling complex joint state-action distributions.

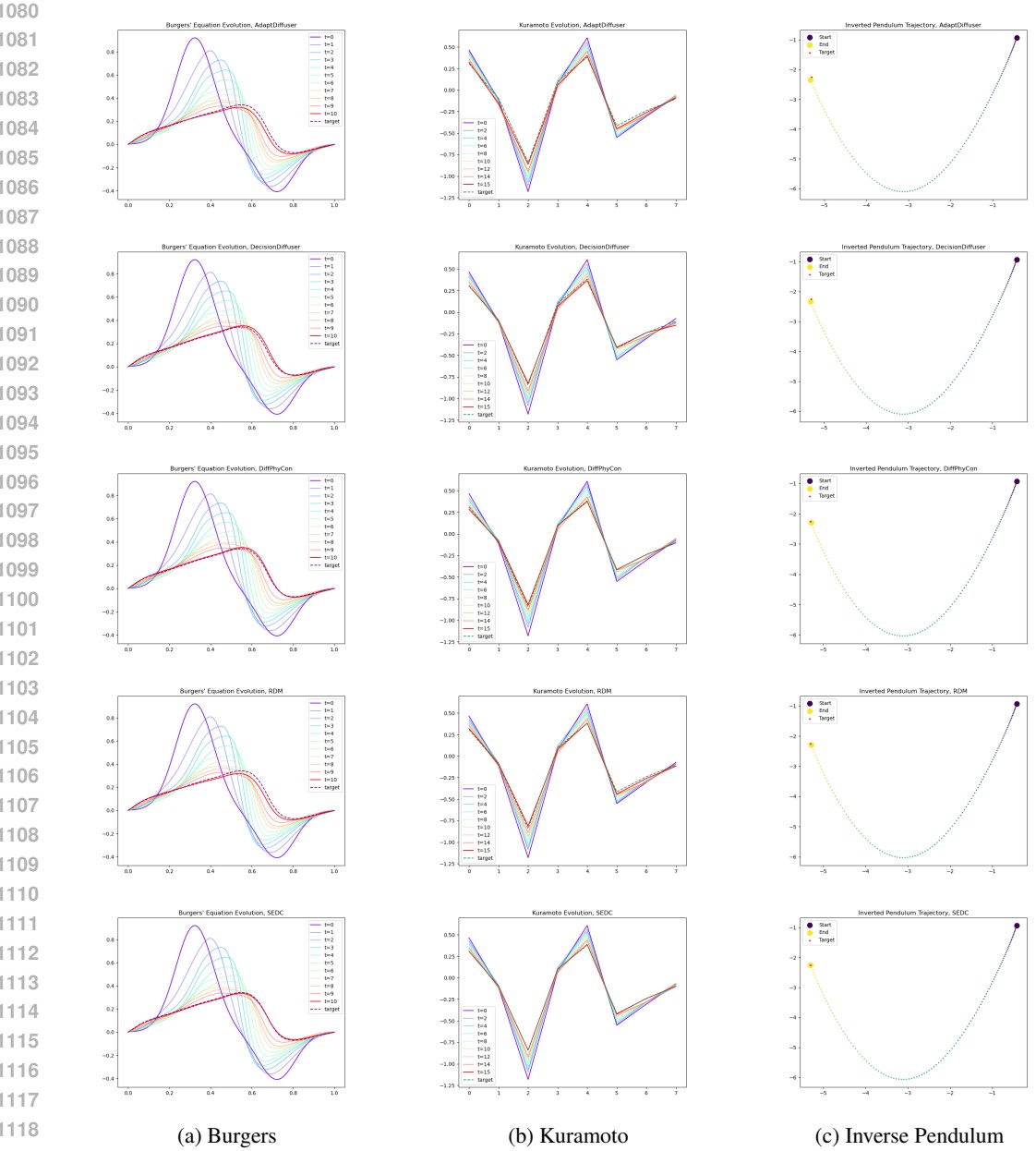

(a) Burgers       (b) Kuramoto       (c) Inverse Pendulum

Figure 8: Comparison of different methods on Burgers, Kuramoto and Inverse Pendulum systems

## G  VISUALIZATION

We present some visualization results of our method and best-performing baselines under three systems. The goal is to make the end state (T=10 for Burgers and T=15 for Kuramoto) close to the target state. As can be seen, SEDC's final state always coincides with the target state. In contrast, the baselines showed inferior results, as some mismatch with the target state can be observed.

## H  LIMITATIONS

In our paper, we assume full state observability throughout. We hope to extend our framework to partial-observable or partial-controllable circumstances in the future.

Considering extending this work to stochastic control settings, our present work focuses on deterministic non-linear systems where SEDC demonstrates significant advantages in sample efficiency and control accuracy. Although we believe the diffusion-based nature of our approach provides a conceptual foundation that could potentially be adapted to stochastic settings, this would require substantial theoretical modifications to our framework components (DSD, DMD, and GSF). Extending to stochastic control would involve addressing additional complexities in modeling state transition probabilities and optimizing over distributions rather than deterministic trajectories. This remains an open research question we are interested in exploring.

While our current framework generates open-loop trajectories, its computational efficiency and ability to produce high-quality plans make it an ideal candidate for integration into a closed-loop MPC scheme. Such an integration would leverage SEDC as a powerful trajectory planner at each step, enabling robust adaptation to unexpected disturbances by replanning in real-time. This synergy between global planning and reactive feedback represents a promising avenue for future research in robust, sample-efficient, data-driven control.

## I USE OF LLMS

The authors used Gemini(Google) to assist with specific formatting tasks and language editing in this work. Specifically, Gemini was used to:

- Convert pre-existing tabular data into LaTeX table format for presentation purposes
- Provide grammar correction and language polishing of author-written text

No content generation, analysis, interpretation, or creation of new ideas was performed using generative AI tools. All original research, methodology, results, and conclusions presented in this work are entirely the authors' own. The use of Gemini was limited to technical formatting assistance and language refinement, similar to using grammar checking tools or word processing software.

