# OpenReview forum: "Sample-efficient diffusion-based control of complex nonlinear systems"
_ICLR.cc/2026/Conference — Submitted to ICLR 2026_

### Official Review · Reviewer_wVxQ · 2025-10-23

**Soundness:** 3
**Presentation:** 4
**Contribution:** 3
**Rating:** 6
**Confidence:** 4

**Summary:**

This paper introduces SEDC (Sample-Efficient Diffusion-based Control) to solve the sample inefficiency of diffusion-based control for nonlinear systems with limited, non-optimal data. SEDC combines three innovations: (1) Decoupled State Diffusion (DSD), which diffuses only on state trajectories (reducing dimensionality) and uses a separate inverse dynamics model to infer actions; (2) Dual-Mode Decomposition (DMD), a dual-UNet denoiser imposing an inductive bias by decomposing dynamics into linear and nonlinear components; and (3) Guided Self-finetuning (GSF), which iteratively refines the policy by augmenting training data with cost-guided trajectories. The insight is that this decoupled and decomposed architecture simplifies learning, while GSF bridges the data-optimality gap. Experiments on systems like Burgers and Kuramoto show SEDC achieves 39.5%-47.3% higher accuracy, matching SOTA performance with only 10% of the training data, with further validation on high-dimensional PDE tasks.

**Strengths:**

- The paper is exceptionally well-written and organized. It clearly articulates three core challenges (dimensionality, nonlinearity, data optimality) and systematically presents three corresponding solutions (DSD, DMD, GSF), making the overall narrative easy to follow.
- The work addresses a significant and practical bottleneck in data-driven control: the high sample cost of diffusion models, especially when trained on limited, suboptimal offline data. The problem analysis is both insightful and highly relevant.
- The proposed methodological innovations, particularly DSD and DMD, are well-motivated and provide strong inductive biases. Decoupling state diffusion (DSD) to reduce complexity and decomposing dynamics (DMD) to handle nonlinearity are both elegant and effective architectural contributions.
- The framework is supported by exceptionally thorough and convincing experimental validation. Beyond demonstrating high sample efficiency in the main results, the paper includes extensive validation on high-dimensional PDE tasks, and noisy observations, which strongly supports the robustness and scalability of the claims.

**Weaknesses:**

- The Guided Self-Finetuning (GSF) component appears to rely on access to an interactive ground-truth simulator to generate new state trajectories for finetuning. This assumption seems to contradict the initial motivation of learning from a purely fixed, offline dataset, blurring the line between offline and online learning.
- The inverse dynamics model in DSD learns a deterministic "average" policy (the conditional expectation) for a given state transition. This approach may be insufficient or suboptimal for complex systems with true multi-modality, where multiple, distinct control actions (e.g., with very different costs) could produce the same transition.
- While effective and validated in ablations, the conceptual novelty of the GSF component is less significant than the architectural innovations of DSD and DMD. It largely follows an established paradigm of iterative self-training, making it more of a procedural refinement than a foundational contribution.

**Questions:**

1.  Does the GSF module require an interactive simulator for finetuning, or can it operate in a purely offline setting?
2.  How does DSD's "average" policy handle control multi-modality, where averaging distinct valid actions (e.g., high-cost vs. low-cost) could result in a suboptimal or invalid action?
3.  Why is the 2nd-order DMD approximation so robust for higher-order nonlinearities, and what are its anticipated failure modes as system complexity increases?
4.  Regarding the DSD trade-off: in which system classes (e.g., chaotic, high action dimension) might learning the inverse dynamics model become as sample-inefficient as the original joint diffusion problem?

---

### Official Review · Reviewer_W3tF · 2025-10-29

**Soundness:** 2
**Presentation:** 2
**Contribution:** 2
**Rating:** 2
**Confidence:** 3

**Summary:**

This paper introduces a diffusion-based method to control nonlinear systems, SEDC (Sample-Efficient Diffusion-based Control).
SEDC is developed to address: (1) high-dimensional state-action spaces, (2) nonlinear system dynamics, and (3) how to go from non-optimal training data to near-optimal control.
Their contributions are structured in three components:
- (i) Decoupled State Diffusion (DSD) structuring the diffusion process.
    - (i-1) They train their diffusion process to generate trajectories of states.
    - (i-2) In parallel, they train an inverse dynamics model to get the control inputs.
    - (i-3) They use in-painting to impose both the initial and the final states.
    - (i-4) They add a cost optimization term directly in the denoising diffusion estimate.
- (ii) Dual-Mode Decomposition (DMD), their network architecture is composed of three networks (one linear and two UNets), imposing a structure intended to explicitly separate the linear and non-linear parts of the system.
- (iii) Guided Self-Finetuning (GSF), to go beyond non-optimal training data, they rollout the trained policy by interacting with the environment, and enrich the training dataset with these new trajectories.

They demonstrate the effectiveness of their method with experiments over challenging control tasks (1-D Burgers, Kuramoto and the inverted pendulum). They show that their method requires fewer training samples than pre-existing methods (hence the name SEDC). The experiment section includes an ablation study of each of their components (DSD, DMD and GSF). In the appendix, they provide an expanded analysis of their method, in particular, they evaluate SEDC over harder tasks (e.g. the Jellyfish locomotion) to demonstrate its scalability.

**Strengths:**

- The presentation of the method in Sec 4 is mostly clear, with each component well presented.
- The authors conducted many experiments on SEDC, with a good ablation study and a well-organized appendix section.
- While I'm not familiar with specific network architectures to address non-linearities of complex systems, their Dual-Mode Decomposition seems to be a good contribution, according to their ablation study.
- From their experiments, SEDC seems better than DiffPhyCon, which is designed to address the same problems.

**Weaknesses:**

To me, the paper is not well-placed in the literature, affecting not only the comparison with prior works, but more worryingly, it concerns how the problem is defined, and so how SEDC's components are presented.

- (Major-1) The goal of this paper is to learn a policy to control dynamic systems. The field of policy learning is divided in subfields depending on the type of data used. Behavior Cloning (BC) operates on a dataset of trajectories without associated rewards/costs. Offline Reinforcement Learning (RL) works with a fixed dataset of state-action-reward trajectories. On the opposite, online RL assumes access to the environment.
In this paper, the authors divide the data-driven approaches in three separate types: BC, RL and diffusion-based. But, to my understanding, BC, Offline-RL and Online-RL are fields, assuming different access to rewards and to the environment. While diffusion-based policies are methods that belong to these fields.
SEDC is introduced as a data-driven method, first assuming a fixed dataset, and reporting performances depending on the size of this dataset. But its third component, GSF, consists in collecting new trajectories by interacting with the environment (page 12, line 622). So, it seems to me, SEDC is an offline-to-online method.

- (Major-2) The problem is not well defined. Methods are first evaluated based on their ability to reach a target state $y_f$, but then the authors add an energy cost $J$ minimization term. It becomes unclear what the objective is. Fig 2, SEDC is compared to methods that do not have access to $J$ (as most are BC methods), except for DiffPhyCon. The idea of presenting the Pareto front of these two objectives is relevant to aggregate the results, but it hides the lack of a clear objective. The idea of plotting the Pareto front should have been credited to DiffPhyCon.

- (Major-3) Most SEDC's components are design choices presented as new ideas.
    - 3-1) The idea of using diffusion for control is not new, and while the wording may differ, all diffusion-based methods mentioned seek to train a policy to control a system. The application to physical systems (e.g. Burgers dynamics, i.e nonlinear wave propagation and turbulent fluid flow) is interesting, but the paper contribution is not to be the first to formulate a control problem as a denoising diffusion process.
    - 3-2) DSD, the idea of diffusing only over the state, is a pre-existing popular design choice. DecisionDiffuser [1] does the same, it diffuses a state-trajectory and learns an inverse dynamics model. So the main contribution highlighted in the abstract "Our approach introduces a novel control paradigm by architecturally decoupling state-action learning and decomposing dynamics" was already present in the work mentioned by the authors as the reference for diffusion-based control frameworks...
    - 3-3) While in-painting the target position was not present in the cited works, authors should mention that the first work on diffusion-based policies, by Janner et al [2], already used in-painting (for the initial position). This is also related to remarks (Major-4) and (Minor-5).
    - 3-5) GSF, the idea of rolling out the policy by interacting with the environment, is the simplest version of offline-to-online.

- (Major-4) Diffusion-based control is described as learning to diffuse the whole trajectory at once. While it was the case in the first diffusion-based control works [2], diffusion policies are now mostly used to generate chunks of trajectories [3], predicting the states or actions over a mid-term horizon. Authors should at least mention this, and maybe consider how their method can work with chunks. Diffusing over the whole trajectory is only possible when T is small (10 or 15 for Burgers and Kuramoto) or when the state space is very small (dimension 2 for the inverted pendulum). In particular, it raises the question of how to adapt the target in-painting and how to define the objective.

- (Minor-5) The target inpainting is misleadingly presented as a "hard constraint", but as $\hat y_T$ differs from $y_T$, it is not a hard constraint, it is actually the target loss of the control system.

- (Major-6) Overall, some important papers are missing in the introduction and in the related work section. For diffusion policies, "Diffusion policy" [3] is not mentioned, while being the most cited work in the field. Behavior PPO is oddly the only work cited to represent "RL". First, it should be made clear that "RL" here is actually offline-RL. Second, even when restraining RL to offline-RL, BPPO is not the only relevant approach. In particular, using diffusion/flow matching for offline-RL and offline-to-online is a trendy topic [4,5], and I believe SEDC belongs to these subfields.

- (Minor-7) BPPO is introduced as "Batch PPO", instead of "Behavior PPO" in the introduction, line 041.

- (Concern-8) I'm not familiar with Burgers and Kuramoto dynamics, so it is hard for me to judge the number of trajectories used (20000). But using 90000 trajectories (+ resampling offline-to-online "GSF") for the inverted pendulum, or even 10% of it, 9000 trajectories, does not convince me of the sample efficiency.

- (Minor-9) The background paragraph on diffusion mixes general diffusion mathematical formulations with applications to control.

Overall, I think this paper should be framed as an application of diffusion control to physical systems, not as developing a new diffusion paradigm ("our approach introduces a novel control paradigm").
Upon my understanding, this paper does offline-to-online RL, and therefore it should position itself with respect to that field. In particular, in their experiments, authors compared SEDC to methods requiring less information and so belonging to different fields (mainly BC). Most of the contributions presented as innovations are design choices, and should rather be presented as such, focusing on their applications to new complex systems.

[1] Ajay, A., Du, Y., Gupta, A., Tenenbaum, J., Jaakkola, T., & Agrawal, P. (2022). Is conditional generative modeling all you need for decision-making?

[2] Janner, M., Du, Y., Tenenbaum, J. B., & Levine, S. (2022). Planning with diffusion for flexible behavior synthesis

[3] Chi, C., Xu, Z., Feng, S., Cousineau, E., Du, Y., Burchfiel, B., ... & Song, S. (2025). Diffusion policy: Visuomotor policy learning via action diffusion

[4] Hansen-Estruch, P., Kostrikov, I., Janner, M., Kuba, J. G., & Levine, S. (2023). Idql: Implicit q-learning as an actor-critic method with diffusion policies.

[5] Park, S., Li, Q., & Levine, S. (2025). Flow q-learning.

**Questions:**

- Why are the experiments with the 2D PDE Jellyfish locomotion only presented in the appendix, while being a more challenging system than the ones presented in the main paper ?
- I did not find anywhere in the paper the number of trajectories recollected with GSF (neither in the main paper nor in the appendix). While Fig 5 presents results after 2 rounds, it is not clear how many trajectories are collected per round. Is it only one trajectory per round, as described in the algorithm Appendix-A ?
- Could the authors explain how the guidance done using the gradient of the cost, Eq-3, relates to classifier-based guidance diffusion?

**Details Of Ethics Concerns:**

Small comments that I am now sure where to write.

The link at the end of the abstract leads to a page saying "Thank you for reviewing our NIPS 2025 manuscript".
And the name in the URL refers to a paper rejected last year at ICLR 2025, "DIFOCON" "A Diffusion-based Generative Approach for Model-free Finite-time Control of Complex Systems".
While resubmitting a paper is not a problem, here the authors indirectly partially de-anonymize their submission.
Also, the Google Drive said to contain the pre-collected data seems empty.

Note on the use of LLMs: This review was done and written without the assistance of any LLMs.

---

### Official Review · Reviewer_3SSR · 2025-10-31

**Soundness:** 3
**Presentation:** 2
**Contribution:** 2
**Rating:** 4
**Confidence:** 4

**Summary:**

The paper introduces Sample-Efficient Diffusion-Based Control of Complex Nonlinear Systems, a framework for sample efficient control of nonlinear systems. The paper introduces three seperate techniques towards this end: 1) decoupling state-based planning with diffusion from an inverse dynamics model 2) a technique for decomposing the trajectory generator into linear and nonlinear components and 3) a technique for iterative refinement by rolling out the current policy. The paper applies the framework to several nonlinear systems, including high-dimensional PDE control.

**Strengths:**

$\textbf{Improvement Over Baselines}$: On the baselines considered, the proposed approach achieves substantially lower target tracking losses with a much smaller dataset than baselines.

$\textbf{Novelty}$: The idea of decomposing the state dynamics into linear and nonlinear components within the diffusion backbone in quite intriguing.

**Weaknesses:**

$\textbf{Limited Experimental Evaluation}$: While the application of the approach to PDEs is refreshing, a major weakness of the paper is that it only compares to baselines on 'toy' robotics problems such as the inverted pendulum. Without comparing to baselines on standard benchmark environments (e.g. hopper, humanoid) it's unclear if the improvements from the approach hold in more realistic settings. In short, empirical evaluation is substantially below the bar for acceptance at ICLR.

$\textbf{Related Work and Novelty}$: A quick google search found several important related works which were not adressed in the paper:


"Learning Coordinated Bimanual Manipulation Policies using State Diffusion and Inverse Dynamics Models" (Chen et al, 2025)

"Latent Diffusion Planning for Imitation Learning" (Xie et al, 2025)

These works also explore decoupling state diffusion from actions, but demonstrate this core idea in substantially more scaled up settings. Given these prior works, there's minimal contributions that I believe the ICLR community will value.

$\textbf{Three different ideas, one paper:}$ The three contributions noted above could each have an entire paper written about them. Moreover, these ideas have little to do with one another, and thus a) none of the ideas are discussed with enough detail in the paper and b) it's difficult to understand what's the key thing I should take away from reading the paper.

$\textbf{Fit for ICLR}$: I rather enjoy the notion of decomposing linear and linear and non-linear dynamics, but I do not believe ICLR is the best venue for this paper. I believe a venue such as L4DC will have more appreciation for these ideas. For the ICLR community to take note of these ideas, it really comes down to whether improvements on standard benchmarks can be shown.

**Questions:**

Questions:

- Can you show that the approach yields benifits on standard benchmarks?

- Can you compare to the prior works I noted?

- Why do all these techniques need to be combined together to achieve strong results?

---

### Official Review · Reviewer_GV3U · 2025-10-31

**Soundness:** 1
**Presentation:** 2
**Contribution:** 1
**Rating:** 2
**Confidence:** 4

**Summary:**

This paper considers using diffusion-based methods to control complex nonlinear systems with high sample efficiency and reliable performance. It presents sample-efficient diffusion-based control (SEDC), which is claimed to address three core challenges: high-dimensional state-action spaces, nonlinear system dynamics, and the gap between non-optimal training data and near-optimal control solutions.

**Strengths:**

1. The structure of the paper is clear.

1. Empirically, the proposed method works better than the baselines.

**Weaknesses:**

1. The paper's technical writing can be improved. The use of the notations is not fully clear to me. Some examples:
    - Line 53: the definitions of $y^P$ and $u^M$ are not clear.
    - The $P$ in Line 53 and the $P$ in Line 133 seem different.
    - Section 3.1 introduces that the dataset $D = \\{\mathbf{u}^{(i)}, \mathbf{y}^{(i)}\\}_{i=1}^P$, but Line 190 says that $\mathbf{x}, k, \mathbf{y}_0^*, \mathbf{y}_f$ are sampled from the dataset. Seems that the dataset does not contain all of them. Also, the definition of $\mathbf{x}$ is unclear.
    - Line 155, $\mathbf{x}^0$ is used to denote the clean trajectory, but in Line 206, the superscript $0$ is used for the final denoised output from the diffusion model, which is different from the clean trajectory from the dataset.
    - Line 258: $B$ is not defined.

1. Many claims in the paper are not supported by evidence or citations. When talking about the drawbacks of existing works, the paper usually only cites the existing work and states the drawback, without mentioning which paper has discussed this drawback. Some examples:
    - Line 53: "This joint distribution implicitly encodes system dynamics of state transitions under external control inputs, which often leads to physically inconsistent trajectories when training samples are insufficient."
    - Line 60: "learning effective control policies from limited data remains particularly challenging for complex systems with strong nonlinearity, such as fluid dynamics and power grids."
    - Line 103: "Supervised learning and reinforcement learning offer adaptive approaches but can also struggle with long-horizon credit assignment and compounding errors."
    - Line 201: "Jointly modeling the state-action distribution is highly sample-intensive and risks generating physically inconsistent trajectories."
    - Line 249: "the limitations of single-network approaches that struggle to model both simultaneously"
    - Line 255: "modeling a dominant linear part and a subtle nonlinear correction is a more stable and sample-efficient task than forcing a monolithic network to learn the entire complex function from scratch"

1. The report of the experimental results lacks error bars in the figures and standard deviation in tables, which are essential for observing the statistical significance.

1. The paper claims to address "high-dimensional state-action spaces", but the proposed experiments are low-dimensional with simple dynamics (I cannot see why the Burgers dynamics have 128 states).

1. The ablation study of the effectiveness of DMD can be improved. The current experiment compares using only the linear intermediate output of the denoising network and the original nonlinear output, but this cannot support the claim that "single-network approaches struggle to model both simultaneously." To support this claim, experiments should be designed to compare the current method with using one network to learn $\mathbf{O}_1 + \mathbf{O}_2$.

1. The RL baseline chosen by the authors is not based on diffusion policies, which makes it vague that whether the performance improvement is from the proposed framework or the use of the diffusion model.

1. The proposed GSF method seems standard in online fine-tuning for BC policies.

**Questions:**

1. This paper seems highly related to offline reinforcement learning. What are the differences? Why are none of the offline RL works discussed?

1. Why does the paper propose to generate the whole trajectory instead of a feedback control policy?

1. Section 4.1, the paper claims to address the risks of generating physically inconsistent trajectories. How can the proposed DSD method do this, given that the inverse dynamics model is learned, and no physical constraints are considered when generating the state-only trajectories?

1. How many seeds are used in the experiments?

1. Given the Burgers dynamics in Appendix B.2, why does this dynamics have 128 states?

---

### Meta-Review · Area_Chair_CSDU · 2026-01-04

**Summary:**

The paper proposes a sample efficient diffusions control of complex nonlinear systems. I agree with the comments of the reviewers on the limitations and novelty of the work. And for this reason, my recommendation is rejection.  The reviewers have raised several concerns related to the writing styling of the paper, the limited experiments and the lack of novelty.

 In addition, there appears to be an issue with anonymity as pointed out by reviewer W3tF.

**Reviewer Concerns:**

None or reviewer concerns have been addressed by the authors.  All reviewer concerns are outstanding.

**Reviewer Scores:**

I do not expect that any reviewers could have raised their scores.

---

### Decision · Program_Chairs · 2026-01-26

Reject